# Methodological quality and reporting characteristics of anthropometric measurements in studies assessing the nutritional status of children in Ethiopia: A scoping review

Mekdes Tigistu Yilma[1,2]*, Alemselam Zebdewos Orsango[1], Mehretu Belayneh[1], Ingunn Marie Stadskleiv Engebretsen[3]

1 School of Public Health, College of Medical and Health Science, Hawassa University, Sidama, Ethiopia,
2 Department of Public Health, Institute of Health Science, Wollega University, Nekemte, Ethiopia,
3 Centre for International Health, Department for Global Public Health and Primary Care, University of Bergen, Bergen, Norway

* mekdestigistu10@gmail.com

## Abstract

### Background

Accurate anthropometric data is essential for assessing nutritional status. To ensure data quality, careful planning of instruments, training and supervision of enumerators are mandatory. In Ethiopia, where malnutrition rates are high, investigating the methodological quality of anthropometric measurements in primary studies is crucial for ensuring the credibility of reports. Therefore, this review assesses the reporting characteristics and methodological quality of anthropometric measurements in primary studies conducted in Ethiopia, focusing on the nutritional status of school children.

### Methods

A comprehensive systematic search was conducted to include primary studies that reported on children's growth from Medline, AJOL, Embase, and CINAHL. Additional sources, such as Google Scholar, ProQuest, Addis Ababa, and Jimma University's repositories were also accessed. Studies whose primary objective was to evaluate the nutritional status of children using anthropometric measurements were included in the review. The results were organized in EndNote, screened in Covidence, extracted in Excel and analyzed in Stata.

### Results

Of the 678 retrieved records, 30 (n = 18,059) studies were included in the review. The studies used different exclusion criteria: 14 (46.7%) excluded children with physical deformities, seven (23.3%) excluded children who received anti-parasitic treatment within a specified time and four (13.3%) excluded children who were

**Data availability statement:** All relevant data are within the paper and its Supporting Information files.

**Funding:** The author(s) received no specific funding for this work.

**Competing interests:** The authors have declared that no competing interests exist.

taking nutritional supplements. About 22 (73.3%) studies did not mention standardization, while 17 (56.7%) did not mention the calibration of instruments. Furthermore, about 12 (40%) studies did not report the setting where the measurements were obtained.

### Conclusion

Inconsistencies in reporting key methodological details of anthropometric measurements were identified, highlighting a potential gap or loose interpretations of the STROBE-nut reporting checklist for studies that measure anthropometry to assess the nutritional status of children. Therefore, we recommend strengthening the STROBE-nut by giving emphasis to the quality assurance aspect of anthropometric measurements including standardization, calibration, material, setting, number of measurements taken and measurer qualifications.

---

### Introduction

Anthropometric measurements are used to indicate the nutritional and health status, dietary adequacy, growth and development of children. These measurements include weight, head circumference, skinfold thickness, height (or length for children under two years or those unable to stand), mid-upper arm circumference and body mass index [1,2]. Accurate measurement of these indicators is essential for understanding the nutritional status of children. However, these measurements could be biased in a situation where there are physical deformities, equipment issues, such as a malfunctioning weight scale, stadiometer, tape meter and skinfold caliper, and skill gaps among individuals taking measurements [1,3].

Anthropometric measurement is a simple, safe, cost-effective and non-invasive procedure used to assess children's growth. However, generating valid anthropometric data requires careful planning on the selection, calibration, and maintenance of instruments, as well as the recruitment, training, standardization, and supervision of enumerators. Therefore, the credibility of the data depends on the thorough reporting of all procedures and quality control activities [4]. A recent study suggested that dietary and food-based measures should supplement anthropometry for measuring child nutritional status, as anthropometry alone may not fully capture the dietary adequacy and clinical and biochemical status of children [5]. Additionally, potential random errors in the methods and procedures for obtaining anthropometric data can affect the accuracy of growth estimates [4].

Malnutrition, particularly undernutrition and obesity, is a global public health concern [6]. In recent years, the coexistence of undernutrition and overnutrition has become increasingly prevalent in low- and middle-income countries [7], including Ethiopia [8]. In Ethiopia, one in five schoolchildren are stunted or wasted [9], while a quarter of children from food-insecure households are overweight/obese [10]. However, the reliability of these findings may be affected by the quality of the anthropometric measurements. Evidence suggests heterogeneity in the quality of

anthropometric data from demographic and health surveys of different countries worldwide, which can compromise the accurate estimates of malnutrition at the population level [11].

A study conducted in Ghana reported that length measurements could be underestimated due to a lack of confidence among measurers and overburden or increased number of measured individuals [12]. Another study also reported that a significant number of anthropometrists in limited-resource settings attain only a satisfactory level of precision in anthropometric measurement after training and standardization [13] which may have implications for the estimate of malnutrition [11].

In a setting where malnutrition is highly prevalent, researchers must ensure the accuracy and reliability of measurements during fieldwork. It is also the researchers' responsibility to report all necessary quality control approaches and methodologies used in assessing anthropometric measurements to enhance the validity of the results. However, only a few studies examined the methodological quality of anthropometric measurements. Therefore, this review assessed the methodological quality and reporting characteristics of anthropometric measurements in primary studies conducted in Ethiopia, focusing on the nutritional status of schoolchildren.

## Methods

### Search strategy

The review was based on the primary studies conducted in Ethiopia from 1 January 2020–13 June 2024 and carried out based on the Preferred Reporting Items for Systematic and Meta-Analysis-Extension for Scoping Reviews guideline (S1_ File) [14]. The search was performed in Medline, AJOL, Embase and CINAHL using the keywords with Boolean operators as *nutritional status OR protein energy malnutrition OR severe acute malnutrition OR nutritional assessment OR anthropometric measurement OR malnutrition OR undernutrition OR overnutrition OR underweight OR overweight OR obesity OR wasting OR thinness OR stunting AND school children OR 5–10 years OR older children AND Ethiopia in all fields and MeSH terms*. However, the search was customized for each of the databases. Furthermore, it was supplemented by Google Scholar, ProQuest, Addis Ababa and Jimma University's repository searches, considering them as the sources of unpublished studies (S2_File).

### Eligibility criteria

Studies whose primary objective was to evaluate the nutritional status of children measured by anthropometric data, conducted in Ethiopia and published between 1 January 2020 and 13 June 2024 were included in the review. To maintain the alignment of the method with the objective of the review, studies that included nutritional status (undernutrition or overnutrition of macronutrients) as an explanatory variable, secondary data sources, focused on the treatment outcome of malnutrition, and studies for which their full text is unavailable (up to 13 June 2024) were excluded from the review.

### Study selection, data extraction and synthesis

The database search results of articles from all databases were recorded, exported to EndNote and stored in a respective database labelled group. Two independent reviewers (MTY and AZO) screened records using Covidence Systematic Review management software based on eligibility criteria by title/abstract and full-text revision after duplicates were automatically removed. Any discrepancies were resolved by a third reviewer (MB) and through discussion.

Data were extracted from included studies in Excel using a pre-structured sheet by MTY, focusing on the characteristics of the studies (study design, study year, region, sample size, objective, population, health status of children, inclusion and exclusion criteria, type of malnutrition (dependent variable) and conclusion. Furthermore, anthropometric measurement-related data were extracted based on the recommended good epidemiological practice that determines the quality of anthropometric measurements, such as calibration method, standardization procedure, measurement qualification,

setting, equipment used and number of measurements [15,16]. The full article and supplementary files were reviewed to extract the methodological detail. We extracted the mean age from studies that did not report the age range in the inclusion or results sections.

The data extracted in Excel were imported into Stata version 17 for analysis. After cleaning for incomplete data by cross-checking the article and data structure, a descriptive analysis of frequencies and percentages was conducted using the number of studies as a denominator.

## Results

### Search results and study selection

A total of 678 articles were found, including 664 from the databases and 14 studies were found through other methods: nine through citation searches and five from university repositories. Of these, only 30 articles fulfilled the inclusion criteria and were included in the review from 65 studies reviewed in full texts, while the remaining 35 studies were excluded for various reasons, as presented in Fig 1.

### Study characteristics

All 30 studies (n = 8,059) included in this review used a cross-sectional study design. Of which, 23 (76.7%) studies were conducted on healthy school children and seven (23.3%) on children living with a medical condition such as epilepsy [17], HIV [18,19], congenital heart disease [20,21], cardiac disease [22] and admitted to an intensive care unit [23]. The majority 13 (43.3%) of included studies were conducted in the Amhara National Regional State [17,18,23–33].

Exclusion criteria's varied across the studies where 14 (46.7%) studies excluded children with physical deformities [10,19,24,25,31,32,34–39], 11 (36.7%) excluded critically ill children [10,19,21,23,32,34,36,37,39], seven (23.3%)

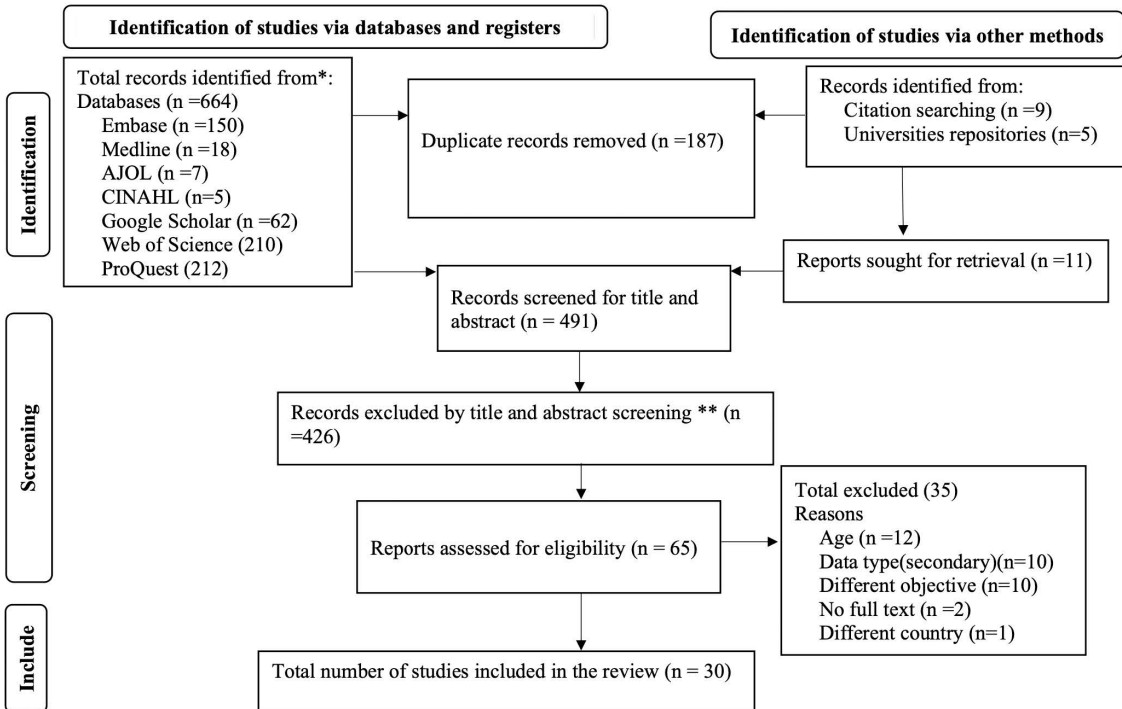

**Fig 1. Flow diagram showing the number of screened, evaluated and included studies from different sources.**

excluded children who received anti-parasitic treatment within a specified time [25–27,34,38] and four (13.3%) excluded children who received nutritional supplements [26,27,34]. As the reasons for and sociodemographic characteristics of refusals could differ from the participants, excluding them could affect the representativeness and magnitude of the problem [40]. However, some studies excluded population samples that refused to participate in the study [26,27,34,37,41]. Nonetheless, measuring anthropometry requires a certain position and there is a possibility of finding a sick child during the study. Studies on diseased children and patients admitted to the intensive care unit did not report how they measure nutritional status and anthropometric measurements of quality control [17–23] (Table 1).

## Anthropometric measurement-related findings

One study assessed nutritional status using the Communicable Disease Control growth chart for children and analyzed the data using WHO Anthro Plus software [24]. The study employed a mixed approach without clear explanation, highlighting a methodological weakness that affects the interpretability of the findings. Regarding the recommended epidemiological practice, 12 (40%) studies did not specify the setting where the measurement was taken [25–28,31–33,35,37,39,41], eight (26.7%) studies did not specify the equipment's used [17–19,22,23,29,35,42] and three (10%) studies did not report the qualification of the measurer [26,32,39]. Moreover, only three (10%) of studies [25,34,38] reported training on anthropometric measurements, while most studies reported provision of training on the objective of the study, content of the tool and approach of data collection without specifically addressing anthropometric measurements.

Weight, height and mid-upper arm circumference were the common anthropometry measurements for primary school children aged 5–10 years. Fourteen (46.7%) studies took two measurements and used the average values to minimize measurement error, whereas the same number of studies did not report the number of measurements they took.

Calibration is not a one-time process; however, 17 (56.7%) studies did not mention the calibration of instruments. Some studies mentioned utilizing calibrated instruments without enough detail [24,31–34,39,41]. Similarly, the majority 22 (73.3%) of studies did not mention standardization of procedure, and only three (10%) of them reported standardization without detail on the number of samples used, how they took the measurements, how they analyzed the data, reported technical error of measurement and their decision and action following the result [25,34,38]. Almost all studies described the standard position for measuring height (Table 2). A summary of the identified methodological gaps is presented in Fig 2.

## Discussion

In this scoping review, we found inconsistencies in the reporting of anthropometric measurement methods and quality control mechanisms, including the number of measurements, qualification and training of data collectors, setting where measurements were taken, materials used, calibration of anthropometric tools, and standardization of measurers in child nutrition studies from Ethiopia. Furthermore, there were variations in the exclusion criteria, such as children with deformities, nutritional supplements, and a recent history of anti-parasitic medication or treatment of intestinal parasites. This review identified a significant gap in reporting anthropometric measurement techniques used for those who are unable to stand independently. The lack of detailed information about the assessment methods raises concerns about the reliability and validity of the findings. Although there was insufficient information about how it was measured, almost all studies did not address the limitations regarding the effect of the above-mentioned anthropometric measurement quality control on the findings. Furthermore, the settings in which the measurements were taken were not reported in many studies. The findings of this scoping review are also in line with another school-based scoping review [43].

Anthropometric measurements are the most common, least expensive and non-invasive methods for assessing nutritional status [1,15]. However, it is highly sensitive to measurement errors due to the contextual setting [44]. Such errors could be imprecision, unreliability, undependability, inaccuracy and bias, which can be estimated by technical error measurement, coefficient of reliability and intraclass correlation [45]. Unless it is reported how it is done and the result of those assessment measurement error findings from standardization analysis, it is difficult for readers to judge the reliability of

**Table 1. Characteristics of included studies, a scoping review in Ethiopia (n = 30).**

| Study | Study year | Region | Design | Sample size | Age (yrs) | Studies exclusion criteria | Health status |
|---|---|---|---|---|---|---|---|
| **Adisu Tafari 2023** | 2022 | Oromia | Community based cross-sectional | 551 | 6-14 | Critically ill, children with deformity and disability children with deformity and disability, closed home for two visits and non-respondent | Healthy children |
| **Ahmed 2022** | 2021 | Addis Ababa | School based cross-sectional | 607 | 7-14 | Deformities and <7 years of age | Healthy children |
| **Argaw 2022** | 2021 | SNNPR | School based cross-sectional | 527 | 6-14 | Critically ill, deformed and unknown age | Healthy children |
| **Asrade 2021** | 2019 | Addis Ababa | Hospital based cross-sectional | 269 | <18 | Premature infants, children with a known genetic disorder, and children with other, non-CD chronic illnesses | Children with cardiac disease |
| **Ayele 2023** | 2022 | Amhara | School based cross-sectional | 600 | 7-12 | Deformity and edematous condition | Healthy children |
| **Bazie 2021** | 2019 | Amhara | School based cross-sectional | 341 | 6-17 | Deformities and child who had taken deworming one month prior to the survey | Healthy children |
| **Berhanu 2022** | 2019 | Oromia | Community based cross-sectional | 606 | 5-14 | Children who had difficulty standing steady or straight, children in a wheelchair, seriously ill during data collection, and those who refused | Healthy children |
| **Berhanu 2023** | 2021 | SNNPR | School based cross-sectional | 494 | 5-19 | Known history of health problems | Healthy children |
| **Biadgilign 2021** | | Addis Ababa | School based cross-sectional | 632 | 5-18 | Children who are permanently ill, have deformities and whose mother is in a morbid state. | Healthy children |
| **Bisetegn 2023** | 2021 | Amhara | School based cross-sectional | 450 | 6-17 | Children who took antiparasitic drugs in the past 4 weeks and nutritional supplements, children who did not volunteer and with other chronic diseases. | Healthy children |
| **Debash 2023** | 2021 | Amhara | Community based cross-sectional | 402 | 6-14 | Children who are taking antiparasitic medication or nutritional supplements and children whose guardian refused to provide consent. | Healthy children |
| **Ayalew 2020** | 2017 | Amhara | School based cross-sectional | 505 | mean age 9 | No specific exclusion criteria were specified. | Healthy children |
| **Geletaw 2021** | 2019 | Somalia | School based cross-sectional | 671 | 6-17 | Treated for intestinal parasite (2 weeks prior) and deformity | Healthy children |
| **Genet 2022** | 2021 | Amhara | Hospital based cross-sectional | 239 | <=18 | Children who had major surgery other than epilepsy surgery | Children with Epilepsy |
| **Hussein 2023** | 2021 | Afar | School based cross-sectional | 936 | 10.96 | children with known disease and refusers. | Healthy children |
| **Molla 2022** | 2021 | SNNPR | School based cross-sectional | 500 | 6-14 | Unknown age, seriously ill and deformities | Healthy children |
| **Teshager 2022** | 2018-2019 | Amhara | Hospital based cross-sectional | 376 | <18 | Patient without a care giver or caregiver with limited information, terminally ill, and on mechanical ventilation | Children admitted to intensive care unit |
| **Tewabe 2020** | 2018 | Amhara | Community based cross-sectional | 392 | 6-12 | No specific exclusion criteria were specified. | Healthy children |
| **Tewabe 2023** | 2018 | Amhara | Community based cross-sectional | 392 | 6-12 | No specific exclusion criteria was specified. | Healthy children |
| **Tiruneh 2022** | 2021 | Amhara | Hospital based cross-sectional | 379 | <15 | Incomplete medical information, children without caregiver and caregiver with mental illness or hearing impairment or medical disorder | HIV-positive children |
| **Tiruneh 2021** | 2021 | SNNPR | Hospital based cross-sectional | 383 | <15 | Incomplete medical information, children without caregiver and caregiver with mental illness, children with physical deformities and seriously ill children. | HIV-positive children |
| **Tsega 2022** | 2020 | Addis Ababa | Hospital based cross-sectional | 228 | <18 | All patients with risk factors other than congenital heart disease that contribute to malnutrition. | Children with congenital heart disease |

*(Continued)*

**Table 1.** (Continued)

| Study | Study year | Region | Design | Sample size | Age (yrs) | Studies exclusion criteria | Health status |
|-------|-----------|--------|--------|-------------|-----------|----------------------------|---------------|
| **Woldesenbet 2021** | 2021 | Addis Ababa | Hospital based cross-sectional | 373 | <15 | Other congenital anomalies and critically ill children | Children with congenital heart disease |
| **Yisak 2021** | 2019 | Amhara | School based cross-sectional | 300 | 6-12 | Visible physical deformities | Healthy children |
| **Bantie 2021** | 2019 | Amhara | School based cross-sectional | 370 | 6-11(+) | Dropouts or absent during the data collection period | Healthy children |
| **Sisay 2022** | 2020 | Amhara | School based cross-sectional | 821 | 6-14 | Seriously ill children and those unable to stand upright by themselves | Healthy children |
| **Eyuel Bekele 2021** | 2021 | Addis Ababa | School based cross-sectional | 350 | 6-18 | New transferred student, who are in the Extension programme are treated with anti-helminths drugs. | Healthy children |
| **Zelalem Destaw 2021** | 2019-2021 | Addis Ababa | Longitudinal | 4500 | 5-19 | Not specified | Healthy children |
| **Abrachew Datiko 2020** | 2020 | SNNPR | School based cross-sectional | 514 | 7-14 | Sick and absent children during data collection time, chest, lower and upper deformities | Healthy children |
| **Etalemahu Mulugeta 2022** | 2021 | Oromia | Community based cross-sectional | 351 | 7-12 | Unknown age, children who are taking antiparasitic drugs, nutrition supplements, and children who are seriously ill or had chronic illness and deformity | Healthy children |

*SNNPR: refers to the South Nation Nationality and Peoples Region.*

the measurements and the findings of the study because imprecise measurement could lead to misclassification in the nutritional status of children [46].

The inherent measurement error of anthropometric measurements can be minimized by calibrating and standardizing of the instrument, procedure and measurer. Calibration is a continuous process that should be done before, during and after taking measurements regularly to examine the functionality of devices and enhance the accuracy of measurements [15]. However, most studies in this review did not report the method of calibration and standardization. In addition to preparing all required instruments and team composition, the demographic health survey also suggests having a standard weight-to-caliber weight scale and conducting a standardization exercise on at least ten samples to enhance the quality of anthropometric data [47].

This review also found that the training of data collectors for standard anthropometric measurements was underreported. Having an anthropometrist is recommended for measuring anthropometry; however, in resource-limited settings like Ethiopia, where anthropometrists are few, it is required to train a measurer to minimize measurement error [15]. Even having anthropometrists with little experience can affect measurement precision [44]. Therefore, training data collectors and field workers on the standard guidelines of anthropometric measurement, continuous calibration of the instrument and securing participants' safety is a critical aspect for improving the quality of anthropometric measurement [47].

The other important factor that determines the quality of anthropometric measurements is the type of instrument used. A simulation study revealed that inaccurate instruments were overestimated the magnitude of obesity [48]. Taking multiple measurements and using average values instead of a single measure also minimize measurement error [45]. Though most included studies did not report the number of measurements taken, a couple of studies even took three measurements to enhance the quality of the data [32,36].

Anthropometry is important for acutely sick children, including those in ICU, to plan care, fluid administration and medication. It can be estimated using various methods, including parental estimate, previous values, measurements at

**Table 2. Study reported characteristics of anthropometry measurements and reporting of limitations regarding anthropometry of included studies, A scoping review in Ethiopia (n = 30).**

| Study | Anthropometry measurement related report | Studys' reported limitation |
|---|---|---|
| **Adisu Tafari 2023** | ■ **Dependent variable**: Stunting and thinness<br>■ **Material**: Weight: digital weight scale; height: no information<br>■ **Calibration**: Weight: prior and before every measurement<br>■ **Standardization**: Done to estimate the Technical Error of the Measurements (TEM) but no result of agreement and further action following the report was discussed.<br>■ **Number of measurements**: two | No reported anthropometry measurement related limitation |
| **Ahmed 2022** | ■ **Dependent variable**: Stunting<br>■ **Material**: Not specified<br>■ **Calibration**: They mentioned as calibrated but did not describe how they calibrate the instrument throughout the field work<br>■ **Standardization**: They mentioned as standardized but did not describe how they standardize the procedure and data collectors.<br>■ **Number of measurements**: two | No limitation reported |
| **Argaw 2022** | ■ **Dependent variable**: Stunting<br>■ **Material**: Tanita HR-200 stadiometer<br>■ **Calibration**: No specific information<br>■ **Standardization**: No specific information<br>■ **Number of measurements**: three | Exclusion of children with deformities may affect the findings. |
| **Asrade 2021** | ■ **Dependent variable**: Undernutrition<br>■ **Material**: not specified<br>■ **Calibration**: No specific information<br>■ **Standardization**: The author mentioned standardize procedure but did not discuss the procedure.<br>■ **Number of measurements**: not specified | No reported anthropometry measurement related limitation |
| **Ayele 2023** | ■ **Dependent variable**: Obesity<br>■ **Material**: Weight: digital balance scale; Height: Stadiometer<br>■ **Calibration**: The weight scale was calibrated at zero with no object on it and placed on the level surface before the measurement was carried out.<br>■ **Standardization**: No specific information except training<br>■ **Number of measurements**: not specified | Skin fold thickness measurement was not done, which might eliminate the limitation of BMI measurement. |
| **Bazie 2021** | ■ **Dependent variable**: Stunting<br>■ **Material**: Digital Seca Germany height scale: a measuring board<br>■ **Calibration**: No specific information<br>■ **Standardization**: Height measurement was standardized using a reference anthropometric measurer before deploying the data collectors to the field.<br>■ **Number of measurements**: two | No reported anthropometry measurement related limitation |
| **Berhanu 2022** | ■ **Dependent variable**: Stunting<br>■ **Material**: Stadiometer with a movable headpiece<br>■ **Calibration**: No specific information<br>■ **Standardization**: No specific information except training.<br>■ **Number of measurements**: two | No reported anthropometry measurement related limitation |
| **Berhanu 2023** | ■ **Dependent variable**: Undernutrition<br>■ **Material**: not specified<br>■ **Calibration**: No specific information<br>■ **Standardization**: No specific information<br>■ **Number of measurements**: two | No reported anthropometry measurement related limitation |
| **Biadgilign 2021** | ■ **Dependent variable**: Overweight/obesity<br>■ **Material**: Weight: SECA digital scale; height: measuring board<br>■ **Calibration**: Measurement was taken after calibration; however, no specific method of calibration was mentioned.<br>■ **Standardization**: No specific information<br>■ **Number of measurements**: two | No reported anthropometry measurement related limitation |
| **Bisetegn 2023** | ■ **Dependent variable**: Undernutrition<br>■ **Material**: Weight: electronic digital scale; height: meter<br>■ **Calibration**: No specific information<br>■ **Standardization**: no specific information<br>■ **Number of measurements**: two | No reported anthropometry measurement related limitation |

*(Continued)*

**Table 2.** (Continued)

| Study | Anthropometry measurement related report | Studys' reported limitation |
|---|---|---|
| **Debash 2023** | ■ **Dependent variable**: Undernutrition<br>■ **Material:** Weight: digital weighting scale; height: vertical meter<br>■ **Calibration:** No specific information<br>■ **Standardization:** no specific information<br>■ **Number of measurements:** two | No reported anthropometry measurement related limitation |
| **Ayalew 2020** | ■ **Dependent variable**: Undernutrition<br>■ **Material:** Weight: electronic scale; Height: wooden stadiometer<br>■ **Calibration:** No specific information<br>■ **Standardization:** They mentioned as standardize procedure (without discussing the specific method, result and action).<br>■ **Number of measurements:** not specified | No reported anthropometry measurement related limitation |
| **Geletaw 2021** | ■ **Dependent variables**: Stunting, and thinness<br>■ **Material:** Weight: digital scale; Height: Seca Rod stadiometer<br>■ **Calibration:** Calibrated tool, but there was not enough detail.<br>■ **Standardization:** The standardization procedure was clearly described with the result. However, they did not mention the method of analysis and software.<br>■ **Number of measurements:** two | Recall bias in the age of children may affect anthropometry. |
| **Genet 2022** | ■ **Dependent variable**: Stunting and wasting.<br>■ **Material:** No information<br>■ **Calibration:** No specific information<br>■ **Standardization:** No specific information<br>■ **Number of measurements:** not specified | No reported anthropometry measurement related limitation |
| **Hussein 2023** | ■ **Dependent variable**: Stunting, thinness and overweight/obese<br>■ **Material:** Weight: digital scale; Height: Seca 217 stadiometer<br>■ **Calibration:** The weighing scale was checked each day prior to the actual data collection. However, there is not enough detail about the calibration of instruments.<br>■ **Standardization:** No specific information<br>■ **Number of measurements:** two | No reported anthropometry measurement related limitation |
| **Molla 2022** | ■ **Dependent variable**: Wasting<br>■ **Material:** Weight: digital electronic scale; Height: Tanita HR-200 stadiometer.<br>■ **Calibration:** Calibration of the weight scale was done using the standard calibration weight of 2 kg iron bars on a daily basis.<br>■ **Standardization:** No specific information<br>■ **Number of measurements:** three | No limitation reported |
| **Teshager 2022** | ■ **Dependent variable**: Wasting<br>■ **Material:** not specified<br>■ **Calibration:** No specific information<br>■ **Standardization:** No specific information<br>■ **Number of measurements:** not specified | No anthropometry related limitation except the use of anthropometric measurement alone to determine nutritional status. |
| **Tewabe 2020** | ■ **Dependent variable**: Undernutrition<br>■ **Material:** not specified<br>■ **Calibration:** No specific information<br>■ **Standardization:** No specific information<br>■ **Number of measurements:** not specified | No limitation reported |
| **Tewabe 2023** | ■ **Dependent variable**: Undernutrition<br>■ **Material:** Weight: digital weight scale (7506 digital scale); Height: a fixed non-bending wooden meter<br>■ **Calibration:** No specific information<br>■ **Standardization:** No specific information<br>■ **Number of measurements:** not specified | No reported anthropometry measurement related limitation |
| **Tiruneh 2022** | ■ **Dependent variable**: Underweight<br>■ **Material:** not specified<br>■ **Calibration:** No specific information<br>■ **Standardization:** No specific information<br>■ **Number of measurements:** not specified | No reported anthropometry measurement related limitation |

*(Continued)*

 

**Table 2.** (Continued)

| Study | Anthropometry measurement related report | Studys' reported limitation |
|---|---|---|
| Tiruneh 2021 | ■ **Dependent variable**: Wasting and stunting.<br>■ **Material:** not specified<br>■ **Calibration:** No specific information<br>■ **Standardization:** No specific information<br>■ **Number of measurements:** not specified | No reported anthropometry measurement related limitation |
| Tsega 2022 | ■ **Dependent variable**: Undernutrition<br>■ **Material:** Weight: weight scale; Height: tape meter<br>■ **Calibration:** No specific information<br>■ **Standardization:** No specific information<br>■ **Number of measurements:** not specified | No reported anthropometry measurement related limitation |
| Woldesenbet 2021 | ■ **Dependent variable**: Undernutrition<br>■ **Material:** Weight: weight scale<br>■ **Calibration:** No specific information<br>■ **Standardization:** No specific information<br>■ **Number of measurements:** not specified | No reported anthropometry measurement related limitation |
| Yisak 2021 | ■ **Dependent variable**: Undernutrition<br>■ **Material:** Weight: digital portable weighting scale<br>■ Height: A height measuring length board<br>■ **Calibration:** A weighing scale calibrated to zero before taking every measurement<br>■ **Standardization:** No specific information<br>■ **Number of measurements:** two | No limitation reported |
| Bantie 2021 | ■ **Dependent variable**: Stunting<br>■ **Material:** Weight: digital scale; Height: wooden measuring board<br>■ **Calibration:** Calibration of the digital weight scale (to zero) and cross-checking it using a pre-known weight material before weighing each pupil.<br>■ **Standardization:** No specific information<br>■ **Number of measurements:** not specified | Measurement bias might also occur, as not having a standardized height measurement. |
| Sisay 2022 | ■ **Dependent variable**: Stunting and thinness<br>■ **Material:** Weight: weight scale; Height: stadiometer<br>■ **Calibration:** The weight scale was calibrated by 2 kg iron bar every day.<br>■ **Standardization:** No specific information<br>■ **Number of measurements:** not specified | No reported anthropometry measurement related limitation |
| Eyuel Bekele 2021 | ■ **Dependent variable**: Stunting and underweight/thinness<br>■ **Material:** Weight: portable digital balance<br>■ **Calibration:** Weighing scales were checked and validated with standard weights every day before the actual weighing of the children.<br>■ **Standardization:** No specific information<br>■ **Number of measurements:** two | No limitation reported |
| Zelalem Destaw 2021 | ■ **Dependent variable**: Malnutrition<br>■ **Material:** Weight: UNISCALE; Height Stadiometer<br>■ **Calibration:** No specific information<br>■ **Standardization:** No specific information<br>■ **Number of measurements:** not specified | No reported anthropometry measurement related limitation |
| Abrachew Datiko 2020 | ■ **Dependent variable**: Undernutrition<br>■ **Material:** Weight: Seca digital scale; Height: stadiometer<br>■ **Calibration:** Calibration was done after each child's measurements.<br>■ **Standardization:** Done using 10 student and done TEM by ENA and it was acceptable.<br>■ **Number of measurements:** two | No reported anthropometry measurement related limitation |
| Etalemahu Mulugeta 2022 | ■ **Dependent variable**: Stunting<br>■ **Material:** Height: a vertical wooden height board<br>■ **Calibration:** The scale is validated by known height and weight and adjusted to zero level after each measurement.<br>■ **Standardization:** states only standard procedure with WHO citation<br>■ **Number of measurements:** two | No reported anthropometry measurement related limitation |

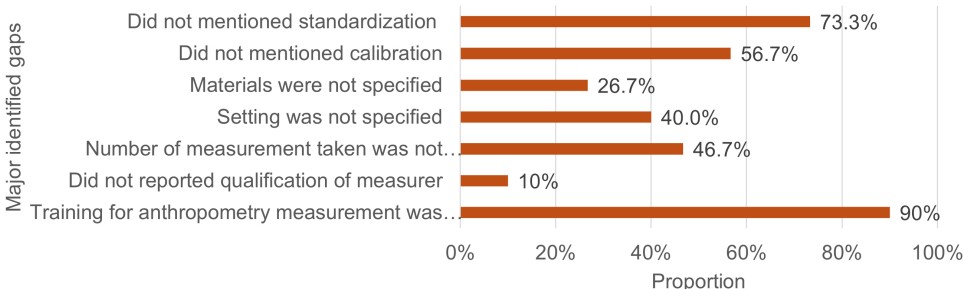

**Fig 2. Summary of major identified methodological gaps of included studies.**

admission, and at regular intervals [49]. Inaccurate measurements in such situations compromise the effect of treatments and hinder individualized care [50]. However, studies conducted in children admitted to the intensive care unit and with other chronic diseases like cardiac disease and epilepsy did not report information about the calibration of instruments and standardization of procedures. Furthermore, most included studies did not report the inclusion or exclusion of children with deformities, acute illness, and casting. However, measuring anthropometry in such conditions may affect the accuracy of measurements [1].

The accuracy of any of the anthropometric measurements significantly affects the child growth indicators; weight-for-height, weight-for-age, height-for-age, and BMI-for-age, as they require at least one anthropometric measurement. Errors in such measurements could result in a systematic error that affects prevalence estimates at population level, trend analysis, determinants and consequences of malnutrition, and targeted program impact [4,51,52]. For example, a child who appears well nourished can be misclassified as malnourished (false positive), resulting in unnecessary strain on the healthcare system. On the other hand, a child who is malnourished may go undetected (false negative), which could prevent them from receiving essential care. Therefore, examining the quality of anthropometric measurement before, during and after data collection is critical for obtaining a precise estimate of nutritional status.

The STROBE extension checklist (STROBE-nut) is available for observational studies evaluating nutritional status, but the checklist lacks sufficient emphasis on ensuring the quality of anthropometric measurements [53]. Although it under point 8 states it as "details of assessment methods", it may be interpreted differently how specific one should be. Therefore, this review highlights the gap in underreporting of the essential aspect of anthropometric measurements, which is required to ensure the reliability and validity of the estimate of nutritional status. For a country like Ethiopia, where resources and infrastructure are limited and the burden of malnutrition is high, this review will provide insight to enhance the quality of the data and to report a significant aspect of anthropometric measurement. However, its main limitation was the inclusion of studies since 2020; however, this may not affect the findings as the review aims to examine the current methodological aspect of anthropometric measurement. Excluding studies that measured anthropometry as an independent variable may affect the findings of this study in different ways. Yet this review provides insights into the need for improving the quality of anthropometric measurements. Exclusion of studies without full text availability upholds the methodological integrity of the review while acknowledging its effect on comprehensiveness. Moreover, inclusion of studies only from Ethiopia also affects the generalizability of the findings. Publication bias could affect the results in either way however, we have minimized its effect through systematic database searches, comprehensive grey literature and citation searches to include all potential published and unpublished studies. Last, but not least, stating what was not found during critical reading may not imply that the methods were not adhered to or reported in adjacent reporting to the published manuscripts. However, if that was the case, a more specific guideline and adherence would have eased reading and interpretation of the work done.

## Conclusion

In this review, we found a considerable gap and inconsistencies in reporting a key methodological detail of anthropometric measurements, which highlights the gap in the reporting checklist for studies that measured anthropometry to assess the nutritional status of children or interpretation issues with respect to depth of reporting. Therefore, we recommend strengthening the STROBE-nut checklist by giving emphasis to standardization, calibration of instruments, materials or equipment used, setting where the measurements were taken, number of measurements taken, and qualifications of the measurers. Furthermore, expanding the review at a regional or global level may provide detailed evidence to the existing body of knowledge and improve the reliability of nutritional status estimates.

## Supporting information

**S1 File. PRISMA-ScR Checklist.**
(DOCX)

**S2 File. Search strategies for different databases.**
(XLSX)

**S3 File. Extracted data generated in this study.**
(XLS)

## Author contributions

**Conceptualization:** Mekdes Tigistu Yilma.

**Data curation:** Mekdes Tigistu Yilma, Alemselam Zebdewos Orsango, Mehretu Belayneh, Ingunn Marie Stadskleiv Engebretsen.

**Formal analysis:** Mekdes Tigistu Yilma.

**Investigation:** Mekdes Tigistu Yilma, Alemselam Zebdewos Orsango, Mehretu Belayneh, Ingunn Marie Stadskleiv Engebretsen.

**Methodology:** Mekdes Tigistu Yilma, Alemselam Zebdewos Orsango, Mehretu Belayneh, Ingunn Marie Stadskleiv Engebretsen.

**Resources:** Alemselam Zebdewos Orsango, Mehretu Belayneh, Ingunn Marie Stadskleiv Engebretsen.

**Software:** Mekdes Tigistu Yilma, Ingunn Marie Stadskleiv Engebretsen.

**Supervision:** Alemselam Zebdewos Orsango, Mehretu Belayneh, Ingunn Marie Stadskleiv Engebretsen.

**Validation:** Mekdes Tigistu Yilma, Alemselam Zebdewos Orsango, Mehretu Belayneh, Ingunn Marie Stadskleiv Engebretsen.

**Visualization:** Mehretu Belayneh, Ingunn Marie Stadskleiv Engebretsen.

**Writing – original draft:** Mekdes Tigistu Yilma.

**Writing – review & editing:** Mekdes Tigistu Yilma, Alemselam Zebdewos Orsango, Mehretu Belayneh, Ingunn Marie Stadskleiv Engebretsen.

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
