## [Decision Letter · Decision Letter 0]

28 Oct 2025

Dear Dr. Yilma,

We look forward to receiving your revised manuscript.

Kind regards,

Ayodeji Babatunde Oginni

Academic Editor

PLOS ONE

Journal Requirements:

2. Please remove all personal information, ensure that the data shared are in accordance with participant consent, and re-upload a fully anonymized data set.

Additional guidance on preparing raw data for publication can be found in our Data Policy (https://journals.plos.org/plosone/s/data-availability#loc-human-research-participant-data-and-other-sensitive-data) and in the following article: http://www.bmj.com/content/340/bmj.c181.long .

4. We are unable to open your Supporting Information file [Supplementary file 3_Data]. Please kindly revise as necessary and re-upload.

Reviewers' comments:

Reviewer's Responses to Questions

**Comments to the Author**

1. Is the manuscript technically sound, and do the data support the conclusions?

Reviewer #1: Yes

Reviewer #2: Yes

Reviewer #3: Yes

2. Has the statistical analysis been performed appropriately and rigorously?

Reviewer #1: Yes

Reviewer #2: Yes

Reviewer #3: Yes

3. Have the authors made all data underlying the findings in their manuscript fully available?

Reviewer #1: Yes

Reviewer #2: No

Reviewer #3: Yes

4. Is the manuscript presented in an intelligible fashion and written in standard English?

Reviewer #1: Yes

Reviewer #2: Yes

Reviewer #3: Yes

Reviewer #1: This manuscript presents a scoping review of primary studies that assessed the nutritional status of children in Ethiopia using anthropometric measurements. The review specifically examines how methodological quality and reporting practices related to anthropometric measurements are described in these studies.

The authors conclude that reporting is inconsistent across the included studies. In particular, many studies failed to specify key methodological details such as the setting in which measurements were taken, whether instruments were calibrated, and whether measurement procedures were standardized.

Based on these findings, the review highlights the need to strengthen reporting guidelines, specifically by recommending that the STROBE-nut checklist be expanded to incorporate quality assurance aspects of anthropometric measurements.

All in all the paper is clear and well written and the conclusion is justified by the findings.

Reviewer #2: See these comments, and recommended article if they would be useful:

Clarify Inclusion/Exclusion Criteria and Rationale

Please clearly justify and elaborate the inclusion and exclusion criteria, particularly exclusion of studies that used anthropometric data as an independent variable and those without full text availability. This is essential to understand potential bias in the representative studies included.

Detail Search Strategy and Reproducibility

Expand on the search strategy details, including exact keywords, Boolean operators, database-specific search strings, and date limits in the Methods section. Providing the full search strings as supplementary material would improve reproducibility and transparency.

Explain Data Extraction and Quality Assessment Procedures

More detail is necessary on how data extraction was conducted, e.g., were forms pilot-tested, how discrepancies between reviewers were resolved, and whether a formal methodological quality assessment or risk of bias was performed on included studies.

Provide Quantitative Summary of Reporting Gaps

While narrative descriptions are given, a more structured presentation (e.g., tables or figures) quantifying missing methodological details (calibration, standardization, measurer qualification) across studies would enhance clarity and impact of results.

Discuss Impact of Reporting Gaps on Nutritional Status Results

The discussion should more deeply analyze how inconsistent reporting and measurement quality potentially biased key nutritional status indicators like stunting and wasting prevalence, with examples if possible.

Include More Detail on Handling Studies with Special Populations

Studies involving sick children or those unable to stand were included but lack details on measurement adaptations. The manuscript should address this critical issue and its implications on data validity.

Address Limitations in More Depth

Limitations related to the focus on Ethiopia only, restriction to recent years, exclusion of micronutrient status studies, and potential publication bias need fuller discussion on how they may have influenced findings.

Strengthen Recommendations with Practical Guidance

The conclusion advocates for checklist improvements but lacks specific actionable recommendations or example items for the proposed quality assurance checklist to guide future researchers.

Language and Formatting Consistency

There are some typographical and formatting inconsistencies (e.g., spacing issues, punctuation, reference citation format) that should be carefully revised to improve readability and professionalism.

Supplementary Materials and Data Transparency

If available, include supplementary files such as the PRISMA flowchart, data extraction templates, and detailed study characteristics in tabular form to enable reader verification and facilitate future research replication.

Recommended article:

What is the carbon footprint of reverse osmosis in water treatment plants? A systematic review protocol

Reviewer #3: The original Abstract percentages appear to be based on an earlier count of n=26 studies, while the Results section correctly uses n=30, making the Abstract factually incorrect.

Location in manuscript Original Text Suggested Correction

Keywords/Abstract/ Main body scooping review scoping review (This appears twice)

Methods, Search Strategy in all field and MeSH terms. in all **fields** and MeSH terms.

Methods, Study Selection after duplicates had automatically removed. after duplicates **were** automatically removed.

Results, Study Characteristics children having different disease conditionslike school children living children having different disease **conditions like** school children living

Results, Anthropometry Except one study [24] all studies defined... Except one study [24]**,** all studies defined... (Missing comma)

Results, Anthropometry training does not specifically address anthropometric measurement; Ensure correct spacing (the line break

General Check and correct the truncated citation at the end of the sentence on page 9: measurements [1, 31.. This is likely a formatting error.

Clarity and Phrasing

Abstract/Conclusion: The recommendation to strengthen the STROBE-nut checklist is a strong conclusion. Ensure the wording is consistent and clear throughout the manuscript (e.g., if there is a 'Discussion' section not provided, this point should be thoroughly developed there).

Results, Anthropometry (Page 18): The sentence "Though they defined it based on the Communicable Disease Control chart, they used WHO Anthro Plus Software for classification” is confusing. It either refers only to one exceptional study or implies a mixed approach. If this sentence refers to the one study [24], it should be clearly linked to avoid the interpretation that all studies used both.

**Do you want your identity to be public for this peer review?** For information about this choice, including consent withdrawal, please see our Privacy Policy

Reviewer #1: No

Reviewer #2: No

Reviewer #3: **Yes:** Abdulmalik Alilu Abubakar

---

## [Author Response · Author response to Decision Letter 1]

25 Nov 2025

Point-by-point response to editor and reviewers

Response to editor

Editor’s feedback

Based on my evaluation of the manuscript, I would like you to improve its grammatical structure to ensure that your study findings are clearly understood by the audience. For example, the statements found in lines 130-138, 148-164, 183-184, and 255-256 need to be rephrased for better clarity.

Response

• Thank you for your valuable feedback. We have carefully addressed the grammatical errors and clarity issues pointed out in lines 130-138, 148-164, 183-184, and 255-256, along with all reviewers’ comments. We have included the captions for supporting information files at the end of the manuscript. We have also reuploaded the extracted data as S3_File in Excel format. The ethics statement also revised as it is specified in the submission system which is N/A for study that does not require ethics statement. The manuscript has been proofread, and we believe necessary corrections have been made to improve the clarity and flow of the text.

Response to reviewers

Reviewer #1

Comment: This manuscript presents a scoping review of primary studies that assessed the nutritional status of children in Ethiopia using anthropometric measurements. The review specifically examines how methodological quality and reporting practices related to anthropometric measurements are described in these studies.

The authors conclude that reporting is inconsistent across the included studies. In particular, many studies failed to specify key methodological details such as the setting in which measurements were taken, whether instruments were calibrated, and whether measurement procedures were standardised.

Based on these findings, the review highlights the need to strengthen reporting guidelines, specifically by recommending that the STROBE-nut checklist be expanded to incorporate quality assurance aspects of anthropometric measurements.

All in all the paper is clear and well written and the conclusion is justified by the findings.

Response

• Thank you for your positive feedback. We appreciate your conclusion that the manuscript is clear and well-written and that you found the conclusions to be justified by the findings. This alignment between our results and their interpretation was a primary goal of our work, and we are happy to learn that the proposed implications are agreeable.

Reviewer #2

Dear reviewer,

• Thank you for your insightful comments.

Comment 1: See these comments, and recommended article if they would be useful:

Clarify Inclusion/Exclusion Criteria and Rationale

Please clearly justify and elaborate the inclusion and exclusion criteria, particularly exclusion of studies that used anthropometric data as an independent variable and those without full text availability. This is essential to understand potential bias in the representative studies included.

Response

• Thank you for raising this point; we have now revised the inclusion and exclusion section. In short, we excluded studies that used anthropometric data as an independent variable mainly to maintain conceptual and methodological alignment with the specific objective of the review. Otherwise, the review could have been very broad, as many studies have some adjustment for nutritional status. Furthermore, if the full text is unavailable or not open access, we could not review the study and extract relevant data, which is a foundation for review (as we don't have funding for paid articles). Exclusion of those studies introduces publication bias and affects the representativeness of the evidence; thus, our conclusion is primarily relevant for online access publications. We also acknowledged the potential limitation in the last paragraph of discussion.

Comment 2: Detail Search Strategy and Reproducibility

Expand on the search strategy details, including exact keywords, Boolean operators, database-specific search strings, and date limits in the Methods section. Providing the full search strings as supplementary material would improve reproducibility and transparency.

Response

• Dear reviewer, thank you for your suggestion, and we agree to the value of sharing the search strategies. However, it could have been very informative if you had specified the gap in search strategy statements because we wrote the search strategies by using exact keywords, Boolean operators and MeSH/Emtree terms as well as the period of relevant findings from January 1, 2020, to June 13, 2024. We have provided the search strategies of three databases in supplementary file 2.

Comment 3: Explain Data Extraction and Quality Assessment Procedures

More detail is necessary on how data extraction was conducted, e.g., were forms pilot-tested, how discrepancies between reviewers were resolved, and whether a formal methodological quality assessment or risk of bias was performed on included studies.

Response

• We appreciate your concern. The form was not pretested because it was pre-specified based on the characteristic of the study and the recommended good epidemiological practice of anthropometric measurement. Regarding discrepancies during the screening process, that was resolved by the third reviewer, and this is also specified in the method (lines 106-107). However, if you mean during data extraction, there was no discrepancy.

• For this scoping review objective, a formal methodological quality assessment (or risk of bias appraisal) is not relevant and typically not a standard or required component of a scoping review. That is why we did not perform a risk assessment. Furthermore, not having a risk of bias assessment is why scoping review is fundamentally different from systematic reviews.

Comment 4: Provide Quantitative Summary of Reporting Gaps

While narrative descriptions are given, a more structured presentation (e.g., tables or figures) quantifying missing methodological details (calibration, standardisation, measurer qualification) across studies would enhance clarity and impact of results.

Response

• Agreed. In addition to Table 1 and Table 2, we have added Fig 2 to summarise the major identified methodological gaps of included studies such as calibration, standardisation, measurer qualifications, number of measurements taken, training for anthropometry measurement, material and setting.

Comment 5: Discuss Impact of Reporting Gaps on Nutritional Status Results

The discussion should more deeply analyse how inconsistent reporting and measurement quality potentially biased key nutritional status indicators like stunting and wasting prevalence, with examples if possible.

Response

• We really appreciate the concern, and we have added a paragraph in the discussion that addresses how inaccurate anthropometric measurements impact the estimate of nutritional indicators like wasting, stunting, underweight and so on with examples.

Comment 6: Include More Detail on Handling Studies with Special Populations

Studies involving sick children or those unable to stand were included but lack details on measurement adaptations. The manuscript should address this critical issue and its implications on data validity.

Response

• Thank you for your suggestion. We believe the point you raised is a relevant research gap. However, providing details on measurement adaptation for inclusion of special populations is out of the scope of this study because this review is about methodological gaps and reporting characteristics in conducted studies. According to our study, studies conducted on children living with medical conditions fail to report all methodological details about how they measured the intended anthropometry, even though it was relevant to ensure the quality of the data.

• If we agreed to suggest a measurement method, it could not be logical because there is no single standard for all special situations. Therefore, suggesting measurement adaptation for special populations requires evaluation of existing methods for each category, like for critically sick children, children with deformities and so on.

Comment 7: Address Limitations in More Depth

Limitations related to the focus on Ethiopia only, restriction to recent years, exclusion of micronutrient status studies, and potential publication bias need fuller discussion on how they may have influenced findings.

Response

• We appreciate your significant concern. It is now revised. However, limitations related to exclusion of micronutrient status studies were deleted from the manuscript because anthropometric measurements are not a direct measure of micronutrient status. Therefore, we believed it was irrelevant and removed it from the text.

Comment 8: Strengthen Recommendations with Practical Guidance

The conclusion advocates for checklist improvements but lacks specific actionable recommendations or example items for the proposed quality assurance checklist to guide future researchers.

Response

• Thank you for reviewing our study in depth. We have recommended strengthening the STROBE-nut checklist by giving emphasis to the identified methodological gaps for studies that measured anthropometry. Therefore, instead of recommending to strengthen the checklist, we have specified the area of improvement for future application.

Comment 9: Language and Formatting Consistency

There are some typographical and formatting inconsistencies (e.g., spacing issues, punctuation, reference citation format) that should be carefully revised to improve readability and professionalism.

Response

• We thank you for your feedback. We have thoroughly revised the entire manuscript to correct all typographical, spacing, and punctuation errors. We have also meticulously checked the reference list to ensure it is complete and consistently formatted according to the Plos style requirements.

Comment 10: Supplementary Materials and Data Transparency

If available, include supplementary files such as the PRISMA flowchart, data extraction templates, and detailed study characteristics in tabular form to enable reader verification and facilitate future research replication.

Response

• We thank the reviewer for highlighting this. All supplementary files were uploaded with our original submission. For example, the flow diagram is presented in Figure 1, study characteristics are described in Table 1, the PRISMA checklist is uploaded as S1_File, the search strategy is uploaded as S2_File, and the dataset used to generate these findings was uploaded as S3_File. However, to ensure they are perfectly clear and easily accessible, we have re-uploaded all supplementary files to avoid any potential technical issues. Regarding the data extraction checklist, as we have extracted the data using a pre-specified Excel sheet, it is the same as the S3_File. So, we didn’t upload it to avoid redundancy.

Comment 11: Recommended article:

What is the carbon footprint of reverse osmosis in water treatment plants? A systematic review protocol

Response

• Dear reviewer, thank you for recommending an article (Abolli, S. et al., 2023). However, we have decided not to incorporate this reference, as it falls outside the scope of our manuscript. Our paper reports on a completed study of methodological quality and reporting characteristics of anthropometric measurement, whereas the suggested study protocol is not a complete study and its focus is on the carbon footprint of reverse osmosis in water treatment plants. Therefore, the aims, methods, and subject matter of the two works are fundamentally different; its topic is not directly relevant to the framework of our research question.

Reviewer #3:

Thank you for providing a critical assessment of our manuscript. We appreciate the opportunity to address the concerns you have raised and to improve the clarity and rigour of our work.

Comment 1: The original Abstract percentages appear to be based on an earlier count of n=26 studies, while the Results section correctly uses n=30, making the Abstract factually incorrect.

Response

• Thank you for identifying this mistake. The percentages in the abstract and result section are now revised and consistent.

Comment 2: Location in manuscript Original Text Suggested Correction

Keywords/Abstract/ Main body scooping review scoping review (This appears twice)

Response

• Thank you for identifying an important aspect. We have removed scoping review from the keywords.

Comment 3:

Methods, Search Strategy in all field and MeSH terms. in all **fields** and MeSH terms.

Methods, Study Selection after duplicates had automatically removed. after duplicates **were** automatically removed.

Results, Study Characteristics children having different disease conditionslike school children living children having different disease **conditions like** school children living

Response

• We agreed to all of the above three suggested typological errors in the methods and results sections and revised them accordingly. We have revised

• field� fields,

• had� were and

• from “...seven on children having different disease conditions like school children living with...” to ”... seven (23.3%) on children living with a medical condition such as ….”

Comment 4

Results, Anthropometry Except one study [24] all studies defined... Except one study [24]**,** all studies defined... (Missing comma)

Response

• We agreed that the previous statement was confusing, so to resolve the lack of clarity in this statement, which was noted by reviewers 2 and 3, we have implemented a comprehensive revision and it now states, “One study assessed nutritional status using the Communicable Disease Control growth chart and analysed the data with WHO Anthro Plus software (optimal growth) [24]. However, the study employed a mixed approach without clear explanation, highlighting a methodological weakness that affects the interpretability of the findings”. The CDC chart is based on children in the U.S., whereas the WHO is for optimal growth in different contexts. We believed the statement is now improved, clear and easy to understand.

Comment 5: Results, Anthropometry training does not specifically address anthropometric measurement; Ensure correct spacing (the line break

Response: We have checked the specific line referenced (page 12, lines 155-157) and confirmed that the spacing in the phrase 'Except in three (10%) of studies [25, 34, 39], the training does not specifically address anthropometric measurement; instead, most studies provide training on the objective of the study, content of the tool and approach of data collection.' is correct, with single spaces between words and after punctuation. Therefore, no spacing changes were made at this specific location except for typos (provide to provided).

Comment 5: General Check and correct the truncated citation at the end of the sentence on page 9: measurements [1, 31.. This is likely a formatting error.

Response

• Thank you. On page 9, there is only Table 1. However, we have reviewed and updated all citations according to the Plos style.

Comment 6: Clarity and Phrasing

Abstract/Conclusion: The recommendation to strengthen the STROBE-nut checklist is a strong conclusion.

A STROBE extension checklist, specifically designed for observational studies focusing on nutritional status (referred to as STROBE-nut), is now accessible. Nonetheless, it is noted that the checklist does not place sufficient emphasis on the quality of anthropometric measurements, indicating a potential area for improvement in the assessment methodologies used within these studies.

Response

• Thank you for your opinion. We agree with you. We have revised the conclusion and specified the key methodological gaps. For example, in the revised draft, the conclusion in the abstract states, “Inconsistencies in reporting key methodological details of anthropometric measurements were identified, highlighting the gap in the STROBE-nut reporting checklist for studies that measure anthropometry to assess the nutritional status of children. Therefore, we recommend strengthening the STROBE-nut by giving emphasis to the quality assurance aspect of anthropometric measurements including standardisation, calibration, material, setting, number of measurements taken and measurer qualification.”

Comment 7: Ensure the wording is consistent and clear throughout the manuscrip

---

## [Editor Report · Decision Letter 1]

2 Jan 2026

Methodological quality and reporting characteristics of anthropometric measurements in studies assessing the nutritional status of children in Ethiopia: a scoping review

PONE-D-24-54149R1

Dear Yilma,

We’re pleased to inform you that your manuscript has been judged scientifically suitable for publication and will be formally accepted for publication once it meets all outstanding technical requirements.

Kind regards,

Ayodeji Babatunde Oginni

Academic Editor

PLOS One
---

## [Editor Report · Acceptance letter]

PONE-D-24-54149R1

PLOS One

Dear Dr. Yilma,

I'm pleased to inform you that your manuscript has been deemed suitable for publication in PLOS One. Congratulations! Your manuscript is now being handed over to our production team.

Kind regards,

on behalf of

Ayodeji Babatunde Oginni

Academic Editor

PLOS One